# Modelling Energy Transition in Germany: An Analysis through Ordinary Differential Equations and System Dynamics

**Andrea Savio** [1,2,*], **Luigi De Giovanni** [3] and **Mariangela Guidolin** [1]

1    Department of Statistical Sciences, University of Padua, 35121 Padova, Italy; guidolin@stat.unipd.it
2    Interdepartmental Centre for Energy Economics and Technology "Giorgio Levi Cases", University of Padua, 35121 Padova, Italy
3    Department of Mathematics, "Tullio Levi-Civita", University of Padua, 35121 Padova, Italy; luigi@math.unipd.it
*    Correspondence: andrea.savio.3@unipd.it

**Abstract:** This paper proposes the application of a multivariate diffusion model, based on ordinary differential equations, to investigate the energy transition in Germany. Specifically, the model is able to analyze the dynamic interdependencies between coal, gas and renewables in the energy market. A system dynamics representation of the model is also performed, allowing a deeper understanding of the system and the set-up of suitable strategic interventions through a simulation exercise. Such simulation provides a useful indication of how renewable energy consumption may be stimulated as a result of well-specified policies.

**Keywords:** energy transition; multivariate diffusion model; system dynamics; simulation; renewable energy; decarbonization

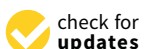



## 1. Introduction

The expression *energy transition* indicates a long-term structural change in energy systems [1]. It is not a new phenomenon: man has made several energy transitions in his short history on Earth. One of the most significant was shifting to an energy system based on fossil fuels (coal, oil, and natural gas) from one essentially based on wood; however, since the 1980s, it has been realized that fossil fuels are at the root of climate change due to carbon emissions into the atmosphere [2]. In short, rapid and deep reductions in greenhouse gas emission are needed to avoid the dangerous effects of climate change and guarantee energy security [3,4]. This transition is often referred to as the decarbonization of the energy sector and aims to shift the system to renewable energy technologies (RETs) by also implying a change from centralized to decentralized energy production. As pointed out in [5], significant reductions can be achieved by using appropriate technologies and policies. Besides a move from fossil fuels to renewable sources, efficiency and savings in the use of energy are seen as complementary viable solutions. In sum, achieving a rapid global decarbonization critically depends on activating contagious and fast-spreading processes of social and technological change within the next few years [6]. According to the International Renewable Energy Agency (IRENA) [7], renewable sources and energy efficiency could potentially reduce carbon emissions from the energy sector by 90%.

With the term *renewable energy*, we normally refer to all forms of energy present on the planet, whose availability is renewed indefinitely over time, unlike fossil fuels [8]. Typical renewable sources available today are hydroelectric, wind, and solar energy, but there are also other sources like, e.g., bioenergy generated by the fermentation or combustion of organic mass (such as fertilizer or plants) and geothermal energy, based on heat coming out from Earth's crust (for example, in the form of a geyser) [9].

In the case of hydroelectric, wind, and solar power, the energy released is typically converted into electricity, although only 25% of electricity produced worldwide comes

from renewables [8]. Indeed, electrification of the energy system will be necessary to fully exploit the potential of renewables and get the energy transition off the ground [10], and forecasting electricity consumption becomes critical for improving energy management and planning by supporting a large variety of optimization procedures. Today, there are significant signs that a regime shift is beginning to occur in many countries [10]. A specific case, analyzed in this paper, is the regime shift in electricity production in Germany. Indeed, Germany has an ambitious plan to realize a transition to sustainable energy with a strong reliance on photovoltaic and wind power for electricity provision [11]. This plan, called "Energiewende" (the German expression for energy transition), was expected to reach 35% of electricity production from renewable sources by 2020 and 80% by 2050. In this sense, the Energiewende is considered the world's most extensive embrace of the wind and solar power [12]. The legal tool behind the Energiewende is the Renewable Energy Act (EEG), promulgated by the German government in 2000. The EEG has favored an exceptional growth in wind and solar energy through the system of *feed-in* tariffs [13]. This mechanism guarantees a minimum purchase price of electricity produced from renewable sources. Three characteristics of German political and cultural history have certainly stimulated the transition: a progressive culture of the environment, legal tools to support RETs, and a historical reluctance towards nuclear energy [14]. The technical aspects of the current energy scenario in Germany within the goals of the Energiewende program are analyzed for instance in [15].

The paradigmatic example of energy transition in Germany may be analyzed focusing on the interdependencies between three energy sources for electricity production, namely coal, natural gas and renewables. Specifically, it is interesting to understand whether renewables have a sufficient competitive strength towards fossil fuels.

Figure 1 describes the yearly consumption time series in Exajoule (EJ) of coal and natural gas between 1964 and 2020 and of renewables (wind and solar jointly considered) from 1989 to 2020 (source: BP Statistical Review of World Energy [16]). As may be easily observed, the time series of coal and gas are simultaneous, while renewables have obviously a more recent history. A possible competitive interplay between these series seems to be captured from the visual inspection of data, but the significance and nature of these effects needs to be tested with a suitable model.

A well-established stream of research has analyzed the temporal dynamics of energy sources by employing multivariate innovation diffusion models. In one pioneering contribution on the topic [17], Marchetti defined energy sources as comparable to commercial products that compete for a market niche, so it is reasonable to analyze the process of adopting a new energy source through diffusion models. Grubler, in [18], highlighted that energy demand and supply systems co-evolve, with technological innovations mutually enhancing each other.

Within the literature produced in the field, in recent times, several contributions have applied bivariate innovation diffusion models to energy contexts, in order to understand the complex dynamics within energy systems: see for instance [1], where the case of Germany is analyzed with reference to the competitive relationship between nuclear power and RETs, [19], which studies the relationship between non-renewable and renewable energy sources in US, Europe, China and India, [20], focusing on the case of energy transition in Australia by comparing the trends of coal, gas and renewables, and [10], where bivariate innovation diffusion models are employed to investigate the specific role of gas in the electricity transition.

Grounding on this literature, this paper presents an analysis of the case of Germany, by proposing a multivariate innovation diffusion model, which considers the dynamic interplay between coal, gas and renewables. This model, which is based on a system of differential equations, is then represented within the system dynamics approach, through the Stock-Flow Network (SFN) tool. Such complementary description encourages a deeper understanding of the relationships within the energy market. The SFN representation aids understanding the complex feedback process acting in the system and creates a suitable

environment for detecting, simulating and evaluating different policy scenarios able to speed up the renewable energy diffusion.

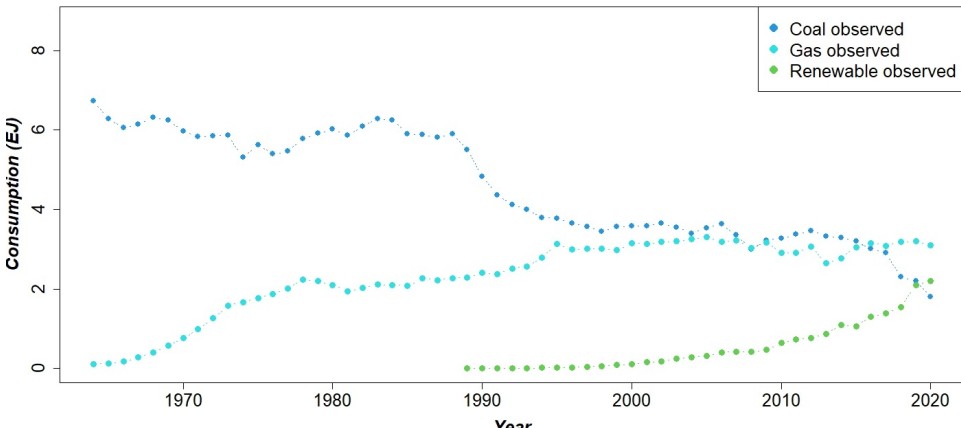

**Figure 1.** Time series of energy consumption of coal, gas and renewables in Germany (1964–2020).

The rest of the paper is organized as follows. Section 2 describes the two methodological approaches employed for model development, respectively based on ordinary differential equations (ODE) and system dynamics. Section 3 describes the Unbalanced Competition among Three Technologies (UCTT) model through its ODE and system dynamics representation, while in Section 4 the model is applied to the case of Germany's energy transition, and a detailed discussion of the results is provided. Based on the results obtained, a simulation study is conducted in Section 5. Section 6 is left for concluding remarks.

Appendix A compares the UCTT model with the 3PM model proposed by [21], while Appendix B discusses some details on the UCTT model.

## 2. Background: Innovation Diffusion Models
### 2.1. Ordinary Differential Equations Approach

The research carried out on diffusion of innovations originated in the 1960s through the definition of the first models based on ordinary differential equations. Pioneering contributions in this regard are, among others, those of [22–27].

The study of diffusion phenomena is a research area that describes and analyzes the growing dynamic of innovations within a social system through a mathematical model [25]. In particular, understanding competition or collaboration dynamics between products or technologies is critical in outlining and describing the trend of diffusion processes [28]. Depending on the situation, the presence of competition can act both as a barrier for the growth of the innovation under consideration and as a stimulus for its development. In the first studies concerning the interactions between several innovations in a single market, researchers used a simplifying assumption to model the time series related to the processes: competition was considered a process in which all the trajectories evolved starting from the same instant in time (*synchronic process*). In [29], the first synchronic model was proposed, which was followed by other works, such as [30,31]. However, in most economic and commercial contexts, products and technologies enter the market at different times. From this consideration, starting from the 2000s, other approaches were developed that consider trajectories starting from different moments in time (*diachronic process*). There are two approaches to analyze competition formalized in *balanced* and *unbalanced* models. Within the balanced models, the same dynamic influences the competitors without distinction. In contrast, [32] were among the first to propose an unbalanced model for competing products by making a distinction to account for different effects within the diffusion of a technology: an internal-influence, which reflects the internal growth dynamics of one technology, and a cross-influence, which identifies the effect of competition exerted by the concurrent technology.

Some important studies that lay the foundations of the methodology used in this work are the balanced model of [33], the mentioned unbalanced model proposed by [32] and the Unbalanced Competition Regime Change Diachronic (UCRCD) model of [34]. Subsequently, the UCRCD model became the basis for some further developments, which include [1,19,20,35,36]. It is important to notice that all the models recalled so far analyze the competition between two products or technologies, while more general models have received a limited attention in literature. To our knowledge, there is just one recent article by [21] proposing a diffusion model for three technologies competing in the same market, the 3PM. The 3PM model has been the starting point for the model proposed in this paper.

### 2.2. System Dynamics Approach

The system dynamics field came to life between the 1950s and 1960s at the Massachusetts Institute of Technology (MIT) thanks to Forrester [37]. This field has been designed to support studying the structure and dynamics of the systems in which people are immersed and to research the answers inside or outside a system. In particular, the primary objective is the design of policies for continuous improvement and the facilitation of implementations and changes. Drawing from the theory of controls developed in engineering and the modern theory of nonlinear dynamic systems, system dynamics often involves the development of formal models and related simulators to capture complex feedback and create an environment for learning and designing system policies [37].

System dynamics allows building the model representing the system under study, identifying the factors of interest, and researching the cause–effect relationships between the internal or external elements of the system [38]. The systems approach complements other approaches by studying interdependencies, allowing synthesizing insights that have been gained through more disciplinary procedures. An essential characteristic of the systems approach, compared to methods that focus on local and immediate effects, is that circular-causality is considered. System dynamics focuses on how interconnected elements affect each other and, through these other elements, themselves again over time [39], thus capturing aspects that are typically beyond the scope of different approaches. A cause–effect relationship can be defined as positive or negative. It is defined as positive when the two events move in the same direction, or as negative when they move in the opposite direction [37]. Even if it may be relatively easy to identify local relationships between pairs of elements, it is normally difficult to evaluate the effects of interventions on the system, due to unexpected or unwanted consequences, system inertia, and counter-intuitive behaviors. Generally, every action generates a reaction reflected on the action itself as feedback: the decision impacts the environment, which in turn affects the decision [40]. The circular-causality relationships cause this feedback, and complex systems are the result of the interaction of multiple feedback. They can be described and analyzed through a formal representation using the Causal Loop Diagram or the Stock-Flow Network's common languages [41]. Our work is inspired by system dynamics analyses that have been applied to electricity generation and energy transition (such as [41–46]). In particular, we develop a Stock-Flow Network (SFN) and present an innovative simulation study based on it. Simulation is an essential tool to quantify the factors that determine the diffusion of renewable energy and propose a "what if" analysis to study some scenarios that could have occurred if the adopted policy in the faced context had been different.

## 3. Materials and Methods

### 3.1. Model Definition with ODE

This section presents the structure of the model for three competitors, according to an ODE representation. The model structure is very similar to the 3PM model proposed by [21], but also presents some crucial differences. A comparison between the two models to underline similarities and differences is illustrated in Appendix A.

Given its peculiar features, the model has been called Unbalanced Competition for Three Technologies (UCTT). The UCTT is characterized by two phases: the first one,

called *competition phase*, where two technologies enter the market at the same time and interact with each other, and the second one, called *double-competition phase*, where the third competitor enters the market and competes against the previous two.

The market potential $m = m_c + m_d$, which is the maximum level of consumption in that market for the analyzed technologies, takes two different sizes: $m_c$, the global market potential under the *competition phase*, and $m_d$, the global market potential under the *double-competition phase*. The model assumes a residual market $m - z(t)$ as a common target for each competitor, with $z(t) = z_1(t) + z_2(t) + z_3(t)$ denoting common cumulative consumption and $z_i(t)$, $i = 1, 2, 3$ indicating the cumulative consumption of technology $i$. The common factor $1 - \frac{z(t)}{m}$ corresponds to the residual market share achieved at time $t$. The model is a system of differential equations where $z'_1(t)$, $z'_2(t)$, and $z'_3(t)$ represent instantaneous consumption of the first, of the second, and of the third technology respectively. The third competitor enters the market at time $t = c_2$, with $c_2 > 0$, creating the two different phases. The indicator function $I_A$ assumes a value equal to one when the event $A$ occurs or zero otherwise.

Parameters of the UCTT are summarized in Table 1, by making a distinction between the *competition phase* and the *double-competition phase*. Parameters referred to as the *competition phase* are indicated with the subscript "*c*", while those referred to as the *double-competition phase* are indicated with the subscript "*d*".

$$
\begin{aligned}
z'_1(t) &= m\{[p_{1c} + (q_{1c} + \delta)\frac{z_1(t)}{m} + q_{1c}\frac{z_2(t)}{m}]I_{t<c_2} \\
&+ [p_{1d} + (q_{1d} + \zeta)\frac{z_1(t)}{m} + q_{1d}\frac{z_2(t) + z_3(t)}{m}]I_{t\geq c_2}\}[1 - \frac{z(t)}{m}] \\
z'_2(t) &= m\{[p_{2c} + (q_{2c} - \gamma)\frac{z_1(t)}{m} + q_{2c}\frac{z_2(t)}{m}]I_{t<c_2} + \\
&+ [p_{2d} + q_{2d}\frac{z_2(t)}{m} + (q_{2d} - \rho)\frac{z_1(t) + z_3(t)}{m}]I_{t\geq c_2}\}[1 - \frac{z(t)}{m}] \\
z'_3(t) &= m\{[p_{3d} + q_{3d}\frac{z_3(t)}{m} + (q_{3d} - \xi)\frac{z_1(t) + z_2(t)}{m}]I_{t\geq c_2}\}[1 - \frac{z(t)}{m}] \\
\\
m &= m_c I_{t<c_2} + m_d I_{t\geq c_2} \\
z(t) &= z_1(t) + z_2(t) + z_3(t)I_{t\geq c_2}
\end{aligned}
\tag{1}
$$

The system outlines the interaction among the three technologies that could result in competition or collaboration. Each phase is characterized by new parameters referred to both the internal and cross-influence for each technology. The internal growth is described through the *seed coefficient* (see Table 1), which represents the initial dissemination of the technology, and the *internal-influence coefficient* (see Table 1), which modulates the technology-specific growth after the innovation phase. Interaction between competitors is outlined through the *cross-influence coefficient* (see Table 1), which measures the effect of the competitors on the consumption of the considered technology in terms of positive or negative influence. Specifically, under the *double-competition phase*, the *cross-influence coefficients* describe the two competitors' effect on a technology. In this case, an assumption is made: that two competitors have the same influence on the third technology (this is a simplifying assumption because it assumes the same effect for two competitors over the third). A negative *cross-influence coefficient* implies a dynamic of *competition*, while a positive one expresses a mutualistic relationship, termed *collaboration*.

**Table 1.** Parameters of the UCTT model (competition phase and double-competition phase).

| | Competition Phase | |
|---|---|---|
| $p_{1c}$ | Seed coefficient of technology 1 | |
| $q_{1c} + \delta$ | Internal-influence of technology 1 | |
| $q_{1c}$ | Cross-influence of technology 2 on technology 1 | |
| $p_{2c}$ | Seed coefficient of technology 2 | |
| $q_{2c}$ | Internal-influence of technology 2 | |
| $q_{2c} - \gamma$ | Cross-influence of technology 1 on technology 2 | |
| | Double-competition phase | |
| $p_{1d}$ | Seed coefficient of technology 1 | |
| $q_{1d} + \zeta$ | Internal-influence of technology 1 | |
| $q_{1d}$ | Cross-influence of technologies 2 and 3 on technology 1 | |
| $p_{2d}$ | Seed coefficient of technology 2 | |
| $q_{2d}$ | Internal-influence of technology 2 | |
| $q_{2d} - \rho$ | Cross-influence of technologies 1 and 3 on technology 2 | |
| $p_{3d}$ | Seed coefficient of technology 3 | |
| $q_{3d}$ | Internal-influence of technology 3 | |
| $q_{3d} - \xi$ | Cross-influence of technologies 1 and 2 on technology 3 | |

### 3.2. Model Definition with System Dynamics

The ODE-based UCTT model may be defined through a System Dynamics approach, with a special focus on the *double-competition phase*, where all the technologies exist in the market. This representation allows us to describe how different system variables are interconnected in feedback structures. In particular, this model has been described as a SFN model and it is reported in Figure 2 (as drawn in its implementation through the simulation software AnyLogic [47]).

The SFN approach allows defining the presence of cause–effect relationships identifying different types of variables: stocks represent state variables that accumulate or deplete over time and thus have a certain level (or state) at a given time; flows are the variables that alter the stocks and detect their variation per time unit; lastly, some auxiliary variables help improve the system readability making the functions that determine the flows' structure more explicit [37].

The system has four stock variables: the residual market, $mRES$, and the three cumulative consumption flows, $z\_1\_t, z\_2\_t, z\_3\_t$, connected by causal links (the flows). These three flows, $z'\_1\_t, z'\_2\_t, z'\_3\_t$, represent the instantaneous consumption variables. The flows' presence explicitly indicates cause–effect relationships that lead to feedback effects, which could be a balancing or reinforcing feedback effect based on the specific case study data. Moreover, several auxiliary variables help to refine the system and are defined as follows. The residual market share, $mSHARE$, which influences the instantaneous consumption variables, is determined by the ratio between the residual market, $mRES$, and the market potential, $mPOT$. The market potential is given by the sum between the residual market and the cumulative consumption of the three technologies, $z\_i\_t$. The cross-influence of each technology is calculated as the difference between its own *internal-influence coefficient* and the respective parameter, $\zeta$, $\rho$, $\xi$. The value of these parameters will be founded on the results of the UCTT model estimation through Nonlinear Least Squares (NLS).

In this way, after identifying the stock and flow system variables, an SFN can be translated into differential equations [37]. Therefore, what distinguishes these models from many other dynamic models is the specification of the equations that outline the modeling process. All non-trivial models with high order and many non-linearities cannot be solved analytically, but numerical integration methods solve the differential equations. SFN models are generally formulated as high-order, non-linear systems translated into stochastic differential equations.

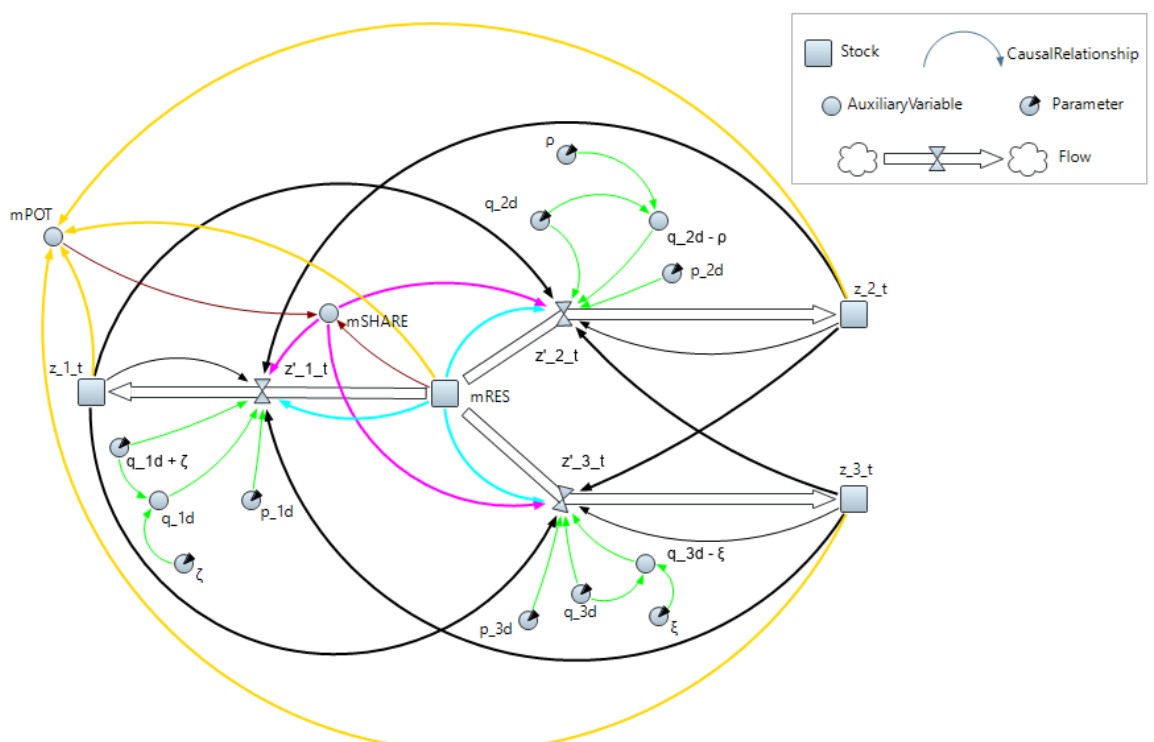

**Figure 2.** UCTT-SFN model. Cause–effect relationships affecting market potential (yellow), market share (brown), coefficients of the technologies diffusion dynamics (green). Cause–effect relationships determined by market share (purple), residual market (light blue), cumulative consumptions on the same (internal-influence, in grey), as well as on different (cross-influence, in black) technologies.

The equations describing the stock variables and the evolution of the system in quantitative terms are:

$$
\begin{aligned}
mRES'(t) &= -(z'\_1\_t(t) + z'\_2\_t(t) + z'\_3\_t(t)) \\
z\_1\_t'(t) &= z'\_1\_t(t) \\
z\_2\_t'(t) &= z'\_2\_t(t) \\
z\_3\_t'(t) &= z'\_3\_t(t)
\end{aligned}
$$

The residual market is depleted by the flows that represent the instantaneous consumption of each technology; meanwhile, these flows, that increase the stocks of the cumulative consumption, are characterized by three differential equations that recall the definitions in Equation (1):

$$
\begin{aligned}
z'\_1\_t(t) &= max\{p\_1d \cdot mRES(t) + (q\_1d + \zeta) \cdot z\_1\_t(t) \cdot mSHARE + \\
&\quad + q\_1d \cdot (z\_2\_t(t) + z\_3\_t(t)) \cdot mSHARE, 0.001\} \\
z'\_2\_t(t) &= max\{p\_2d \cdot mRES(t) + q\_2d \cdot z\_2\_t(t) \cdot mSHARE + \\
&\quad + (q\_2d + \rho) \cdot (z\_1\_t(t) + z\_3\_t(t)) \cdot mSHARE, 0.001\} \\
z'\_3\_t(t) &= max\{p\_3d \cdot mRES(t) + q\_3d \cdot z\_3\_t(t) \cdot mSHARE + \\
&\quad + (q\_3d + \xi) \cdot (z\_2\_t(t) + z\_3\_t(t)) \cdot mSHARE, 0.001\}
\end{aligned}
$$

## 4. Model Application and Results

This section presents the results of the UCTT model to the yearly time series of consumption of coal, gas and renewables (solar and wind) in Germany (Figure 1). The UCTT allows to statistically test the nature and significance of interactions between these

three energy sources. The estimates obtained are then employed in the SFN, in order to detect reinforcing and balancing feedback loops and to structure and implement the simulation study.

### 4.1. UCTT Model Application

This section describes the application of the UCTT model. The model fitting is illustrated in Figure 3.

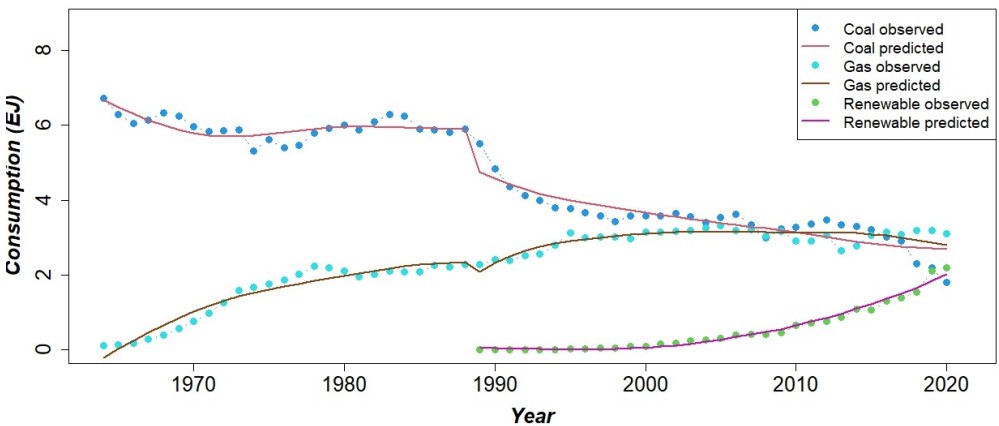

**Figure 3.** UCTT model for coal, natural gas, and renewables in Germany from 1964 to 2020.

The model obtained a satisfactory result in terms of *goodness-of-fit* ($R^2 = 0.9903495$) and significance of parameters as a general outcome. Parameter estimates are outlined in Table 2. As may be observed, despite some slight instabilities in some parameters concerning the *double-competition phase*, namely $p_{3d}$, $q_{2d}$ and $\rho$, the model is well identified and the relationships between the three competitors take a clear meaning. A visual inspection of Figure 3 allows us to appreciate the ability of the model to account for these patterns: renewables exhibit an initial exponential consumption trend that seems to be affecting coal and natural gas trajectories, outlining the decarbonization process. Coal consumption is in a clear decline phase: after an initial period of stability, a sharp decline is evident around the early 1990s; in particular, this is the crucial moment when renewables entered the market. The curve for natural gas also has a particular shape: during the first phase, where it is competing with coal, its consumption is increasing, while it has a more stable structure, around a value of 3 EJ, after the introduction of renewables in the market.

The model allows identifying the main result in terms of "competition-collaboration", by interpreting the *cross-influence coefficients*. These parameter estimates, concerning the *double-competition phase*, when coal, gas and renewables are all in the market, are reported in Table 3.

Analyzing the *double-competition phase*, the *seed coefficients*, $p_{1d}$, $p_{2d}$ and $p_{3d}$ present low values; in particular, for renewables, the weak and not significant value of parameter $p_{3d}$ reflects the initial difficulty of these technologies in entering the market and the subsequent need of incentives to stimulate their initial growth.

After the innovation phase, renewables have been experiencing a strong internal growth, as testified by the high value of the *internal-influence coefficient*, $q_{3d} = 0.3123$. At the same time, the corresponding estimates for coal and natural gas are negative and relatively weak, $q_{1d} + \zeta = -0.0553$, $q_{2d} = -0.0708$, which is coherent with the declining trend of both series. Specifically, the internal-influence parameter for gas $q_{2d}$ is not significant, a reasonable outcome given its stable trend in this phase.

Regarding the interplay between the three technologies, the *cross-influence coefficients* help interpret how the three technologies interplay. Coal and natural gas are characterized by a positive cross-influence effect, i.e., $q_{1d} = 0.0707$ and $q_{2d} - \rho = 0.0936$, which highlights that the competitors *collaborate* with coal and gas; in particular, renewables do not seem to be able to exert a sufficiently strong competitive power. In contrast, renewables are

characterized by a negative, but weak, cross-influence parameter, $q_{3d} - \xi = -0.0014$, which shows that coal and natural gas *compete* with the new entrant technologies, trying to limit their expansion. However, it should be remarked that renewables are characterized by a strong and rapid growth, as already noticed from the value of the internal-influence parameter, $q_{3d} = 0.3123$.

A similar result concerning renewables was evidenced in [1], where the interplay between nuclear and renewables was studied through the UCRCD model. The UCRCD model made it possible to uncover the direct relationship between the nuclear and renewables in terms of internal and cross influence. Specifically, the authors found that renewables exerted a significant and measurable effect in determining the observed decline of nuclear power consumption.

The UCTT model deals with a more complex case, where the action of three energy sources is considered.

### 4.2. UCTT—SFN Model Application

Based on the estimates obtained, it is interesting to analyze the cause–effect interdependencies between the entities and recognize the feedback acting in the system. In particular, after assigning the estimated value found through the UCTT model (see Table 2) to each entity of the SFN model in Figure 2, balancing and reinforcing feedback may be captured. Coal, gas, and renewables are, respectively, referred to elements 1, 2, and 3 of Figure 2. The relevant feedback loops of the SFN for the UCTT model for the German case are reported in Figure 4. These elements will be essential to make dynamic hypotheses on the system evolution and identify where to take action to reinforce or balance specific tendencies.

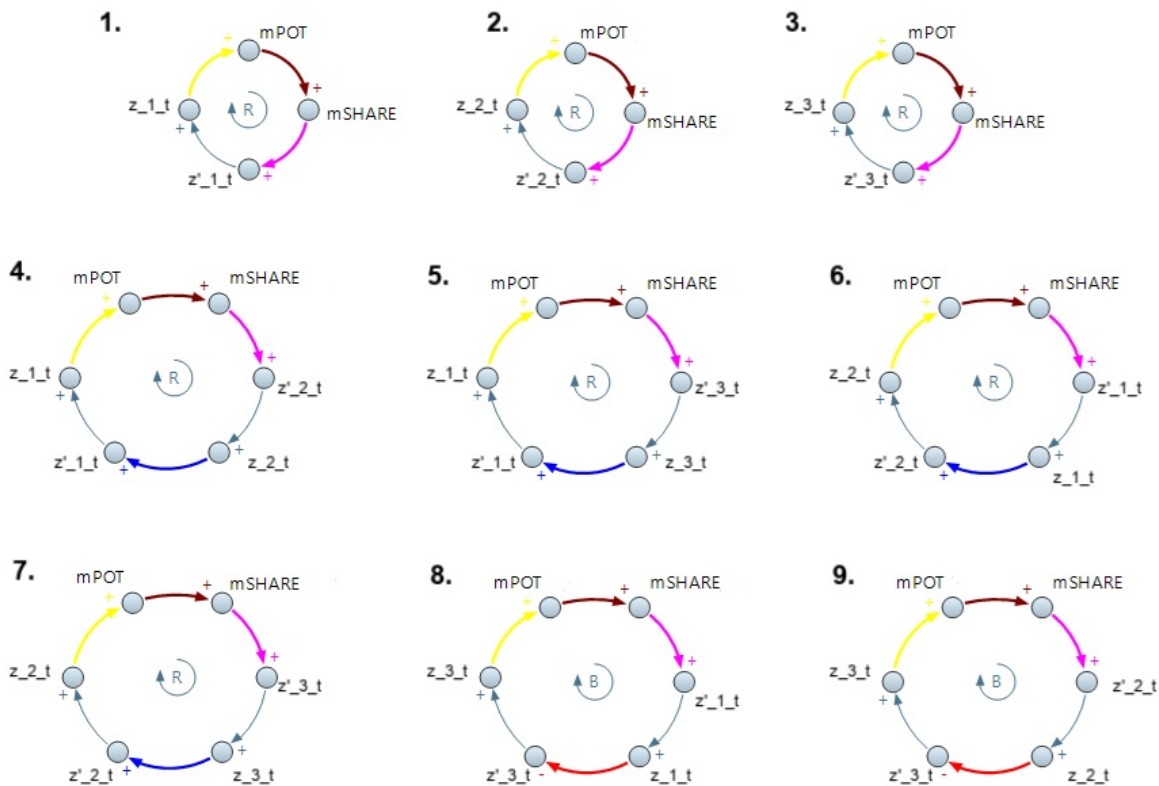

**Figure 4.** UCTT-SFN relevant feedback loops. Cause–effect relationships affecting market potential (yellow), market share (brown). Cause–effect relationships determined by market share (purple), cumulative consumptions on the same (internal-influence, in gray), as well as on different (cross-influence, positive in blue and negative in red) technologies.

The first three feedback loops describe each technology's dynamic as reinforce feedback: an increase of the potential market generates an expansion of the market share, implying an increase of instantaneous consumption that elicits a push on effect-variable, i.e., the cumulative consumption, causing an increase in the market potential.

The following feedback loops are the ones that determine the interaction between the three energy sources. The most critical element is the connector that goes from the cumulative consumption of an energy source to the instantaneous consumption of the other technology (the connectors from z_i_t to z′_j_t, with $i, j = 1, 2, 3, i \neq j$) representing the cross-influence, which defines the relationship between the two competitors. These connectors are colored in blue or red in Figure 4 depending on the parameter estimates in Tables 2 and 3: the blue link defines a positive cause–effect relationship, while the red one underlines the presence of a negative cause–effect relationship.

**Table 2.** UCTT model parameter estimates.

| Parameter | Estimate | Std.Error | Lower c.i. | Upper c.i. | *p*-Value |
|:---:|:---:|:---:|:---:|:---:|:---:|
| $m_c$ | 483.5858 | 116.9050 | 254.4561 | 712.7154 | 0.0002 |
| $p_{1c}$ | 0.0142 | 0.0036 | 0.0072 | 0.0213 | 0.0003 |
| $p_{2c}$ | −0.0009 | 0.0004 | −0.0018 | −0.0001 | 0.0266 |
| $q_{1c}$ | 0.1500 | 0.0417 | 0.0683 | 0.2317 | 0.0008 |
| $q_{2c}$ | −0.0389 | 0.0279 | −0.0936 | 0.0157 | 0.1690 |
| $\delta$ | −0.1694 | 0.0454 | −0.2584 | −0.0804 | 0.0006 |
| $\gamma$ | −0.0775 | 0.0331 | −0.1424 | −0.0126 | 0.0240 |
| $m_d$ | 370.7024 | 23.0794 | 325.4677 | 415.9372 | 0.0000 |
| $p_{1d}$ | 0.0135 | 0.0011 | 0.0113 | 0.0157 | 0.0000 |
| $p_{2d}$ | 0.0048 | 0.0007 | 0.0035 | 0.0061 | 0.0000 |
| $p_{3d}$ | 0.0002 | 0.0003 | −0.0005 | 0.0009 | 0.5960 |
| $q_{1d}$ | 0.0707 | 0.0213 | 0.0290 | 0.1124 | 0.0013 |
| $q_{2d}$ | −0.0708 | 0.0510 | −0.1707 | 0.0292 | 0.1690 |
| $q_{3d}$ | 0.3123 | 0.0423 | 0.2294 | 0.3952 | 0.0000 |
| $\zeta$ | −0.1260 | 0.0391 | −0.2027 | −0.0494 | 0.0018 |
| $\rho$ | −0.1643 | 0.0915 | −0.3436 | 0.0149 | 0.0759 |
| $\xi$ | 0.3137 | 0.0438 | 0.2279 | 0.3995 | 0.0000 |

$R^2 = 0.9903495$
$SSE = 6.622326$

**Table 3.** UCTT model parameter estimates (double-competition phase) and their description.

| | Estimate | Description |
|:---:|:---:|:---|
| $m_d$ | 370.70 | Market Potential |
| $p_{1d}$ | 0.0135 | Coal—Seed |
| $q_{1d} + \zeta$ | −0.0553 | Coal—Internal-influence |
| $q_{1d}$ | 0.0707 | Coal—Cross-influence |
| $p_{2d}$ | 0.0048 | Natural Gas—Seed |
| $q_{2d}$ | −0.0708 | Natural Gas—Internal-influence |
| $q_{2d} - \rho$ | 0.0936 | Natural Gas—Cross-influence |
| $p_{3d}$ | 0.0002 | Renewables—Seed |
| $q_{3d}$ | 0.3123 | Renewables—Internal-influence |
| $q_{3d} - \xi$ | −0.0014 | Renewables—Cross-influence |

In this way, loops 4, 5, 6, 7 represent reinforcing feedback (highlighted with a "R" in the center) and explain how the instantaneous consumptions of coal and natural gas are affected positively by the competitors' consumption in a cooperative manner. According to feedback loops 4 and 5, for example, an increase in natural gas and renewable energy

consumption helps the consumption expansion of coal, and an increase in coal and renewable energy values "reinforces" the diffusion of natural gas. An increase in renewable energy consumption stimulates growth even in gas consumption and vice versa, due to the presence of the reinforcing feedback loop 7. We notice that the cross-influence connector that identifies a positive relationship of renewables over gas plays a key role in this loop.

The last two feedback loops, 8 and 9, present a red arrow that defines a competitive situation for renewable energy. These are balancing feedback loops (highlighted with a "B" in the centre) that point out how the other two technologies hold back the expansion of the renewable source. For example, balancing feedback loop 8 suggests that an increase in coal consumption may even cause renewables' dampening. Furthermore, an increase in renewables consumption will affect coal consumption in turn, as a brake on this spread.

The initial value of the three stock variables, representing the cumulative consumption, is the first available value for each technology series in the double-competition phase (the value observed of yearly consumption in 1989). The initial value of the potential market is calculated by the difference between the $m_d$ value in Table 2 and the first values of the other three stocks. Furthermore, even the auxiliaries variables or parameters values are taken from Table 2. After assigning each entity of this system the correctly estimated value, the SFN, implemented and run in AnyLogic [47], allows us to simulate the double-competition phase case. Figure 5 illustrates the trajectories of the instantaneous consumption of the three technologies. The curves on the time plot are very similar to the predicted ones in Figure 3, since the differential equations that determine the instantaneous consumption are similar to those in System (1). In addition to the time plot, it is provided a bar chart that highlights the EJ cumulative consumption of each energy source: coal and natural gas dominate the market; however, renewable energy is spreading quickly.

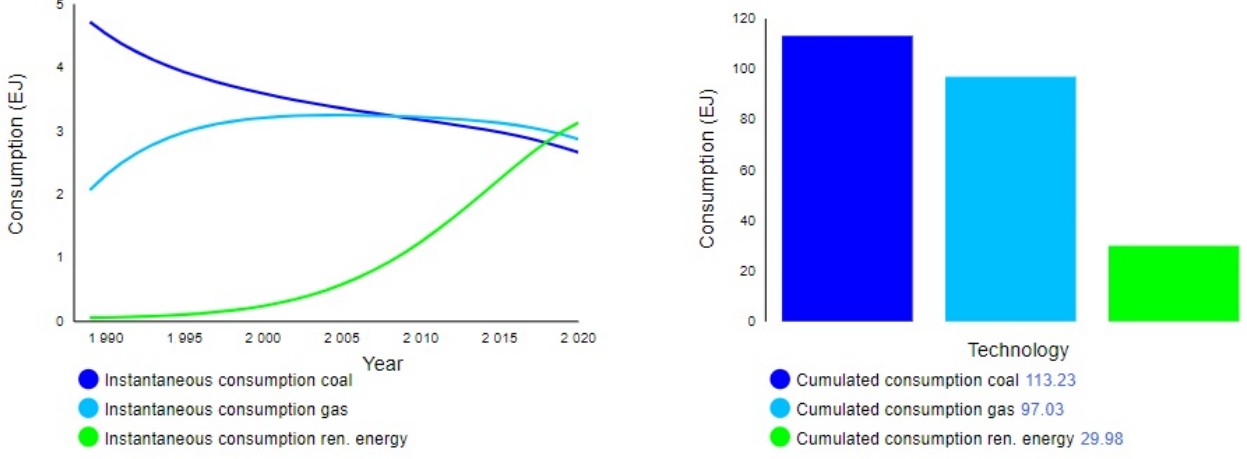

**Figure 5.** Simulation of coal, natural gas, and renewable energy consumption in the observed scenario.

## 5. Simulation Study for Alternative Scenarios

This section analyzes two possible scenarios that Germany could have faced instead of the observed one, by performing different policy choices. It is interesting to discover what would have happened, on the one hand, if the government had set an increase in incentives to renewable energy consumption; on the other hand, if the government had enacted restrictions on consumption of coal and natural gas. The study focuses on acting on specific cause–effect relationships identified by the analysis of the feedback loops presented in Figure 4, as to determine where to intervene effectively with policy actions.

### 5.1. Policy 1: Stimulate Consumption of Renewable Energy

The first proposed simulation study focuses on a crucial aspect of the diffusion of an energy source: incentives. Governments and institutions typically use the incentive tool to stimulate demand for new technologies. In this sense, it is interesting to ask what

would have happened if the German government had-ideally-increased the incentives for renewable energy consumption. This increase would have led to a reduction in the competition of fossil fuels towards renewables. In practical terms, this would have decreased (in absolute value) renewable energy's *cross-influence coefficient* value, which it is represented by connectors {z_1_t - z'_3_t} and {z_2_t - z'_3_t} in balancing feedback loops 8 and 9 of Figure 4. For this reason, a situation has been simulated in which an (absolute) value lower than that previously estimated is assigned to the cross-influence of renewables. Specifically, the hyperparameter $\zeta$ value has been changed from 0.3137 to 0.3127. Consequently, the *cross-influence coefficient* goes from a value of $-0.0014$ to $-0.0004$. This is a substantial decrease (in absolute value), in line with an ideal significant change in consumption. Figure 6 shows the graphical results of this simulation.

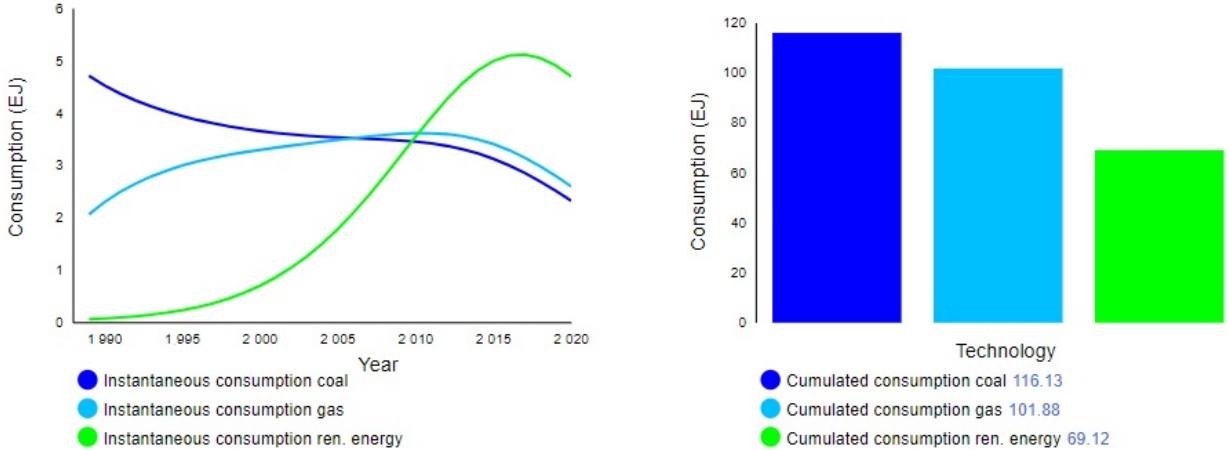

**Figure 6.** Simulation results of Policy 1.

This simulation leads to several interesting conclusions. An increase in incentives to consume renewable energy causes an evident acceleration of the diffusion. In fact, this source reaches a peak of 4 EJ, while, according to the model based on real data, it reached a maximum yearly consumption of 2.2 EJ. Renewable energy outperforms coal and natural gas in this simulated scenario. Coal and natural gas trajectories are not overly modified: they are detected by a declining evident phase, but the curves are similar to those observed.

*5.2. Policy 2: Limit Consumption of Coal and Natural Gas*

Another viable strategy to stimulate the consumption of renewable energy by a government is to impose restrictions on the consumption of competitive sources. In this case, the situation that would have occurred if the German government had imposed restrictions on coal and natural gas use has been simulated. This barrier penalizes the internal growth of the two non-renewable sources, favoring, in this way, the expansion of renewable technology.

In particular, analyzing the observed gas and coal consumption trajectories, a decrease in the value of these two technologies' internal-influence is hypothesized in two different ways, respectively. The consumption of coal is in a phase of decline, so a restriction will have a more significant influence on the negativity of internal-influence compared to the gas source, which has a more constant trend. For this reason, within the simulation, the *internal-influence coefficient* of the coal source is decreased by about 20%, while the one of the natural gas source by 10%. These changes are affecting, respectively, the connectors {z'_1_t - z_1_t} and {z'_2_t - z_2_t} and, hence, all the reinforcing feedback loops of Figure 4 (except loop 3). The graphic result of the simulation is shown in Figure 7.

The decrease in the internal-influence of the two technologies causes some significant changes in the trajectories of the curves. The gas and coal curves show a drop in the level of consumption: in reality, both move around the values of 3 and 4 EJ, while in this simulation,

they stabilize around a value below 3 EJ. The trend in gas consumption remains pretty constant, while the downward trend in coal consumption is more pronounced. Renewable energies benefit from this situation by acquiring ever greater market shares. The expansion has an exponential trend that brings the consumption of this energy source to a peak over the 5 EJ. The observations confirm that a restriction applied to the consumption of competitive sources could have created an advantage for renewable energy technologies.

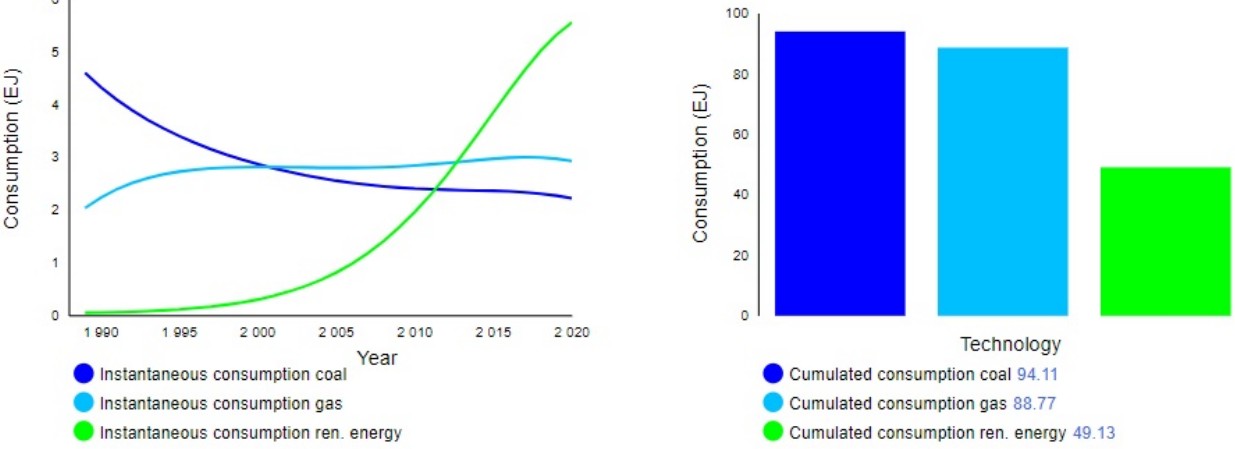

**Figure 7.** Simulation results of Policy 2.

## 6. Conclusions

Achieving sustainable, stable and secure energy systems is reaching the top of many countries' agendas in recent years. Besides climate change problems, the COVID-19 pandemic, with the long periods of lockdowns imposed on populations, has strongly highlighted the critical importance of resilient and affordable electricity systems. Moreover, energy security is increasingly becoming a major issue. At the time of writing this paper, the world's attention is focused on the terrible events occurring in Ukraine. While there is no certainty on how this situation will evolve in the future and the need to find a solution to the conflict is the priority, one point that has become clearer than ever, especially to European countries, is that energy independence from oil and gas is a primary objective in the short-term. The importance of an energy transition, by investing in renewables growth, has become central in the public discourse in the last few weeks.

In March 2022, right after the outbreak of war in Ukraine, the IEA provided a 10-point plan for the European Union, in order to reduce consumption of Russian gas and support the European Green Deal [48]. One of the points foresees an acceleration in the deployment of new wind and solar projects. At the same time, the report clearly states that the process of cut imports of Russian gas will not be simple. Two major aspects will be crucial for a successful implementation according to [48], (i) a concerted and sustained policy effort with a strong dialogue on energy market and security, and (ii) a clear communication between governments, industry and consumers.

The analysis conducted in this paper obviously does not consider these recent events, but it may provide a useful mean to describe and study, in a relatively simplified way, the dynamics occurring between these critical energy sources. We selected the paradigmatic case of Germany which, thanks to the Energiewende, is projected to become a low-carbon, nuclear-free energy system by the middle of the century [15]. Focusing on the interdependencies between renewables and the most used sources for electricity production, gas and coal, the UCTT model indicated that renewables are strongly growing and sustained by a widespread belief of people towards sustainability objectives. This essentially confirms the findings of [1]. On the other hand, the model suggested the presence of a significantly negative effect played by gas and coal on renewables, limiting their expansion, while there appeared to be a kind of "collaboration" between coal and gas, testified by the positive sign

of the respective cross-influence parameters, giving rise to a lock-in situation [10]. This is a different and more complex outcome with respect to that observed in [1], where renewables were found to have a significantly negative strength on nuclear power. However, in [1], it was also acknowledged that Germany was ready to implement policies to accelerate the regime shift favoring renewables at the expense of nuclear consumption well before the Fukushima disaster influence.

This stresses the essential role played by policies, helping Germany in realizing a cost-efficient, fair and sustainable path towards its very ambitious energy transition goals [49]. The SFN approach yielded significant results both qualitatively and quantitatively. The most interesting qualitative result is the direct identification of further feedback compared to the ODE approach: the feedback loops of Figure 4 allowed a preliminary contextual analysis of the cause–effect relationship acting in the system, the related feedback and interplay, as well as their role in determining the dynamic of the energy market. This analysis helps to identify the points where to act in order to reinforce or balance consumptions' trends, and can be conveniently exploited for the preparation of policy scenarios. From a quantitative point of view, the SFN model allowed measuring the cause–effect relationships, which is the base for scenarios simulation. This aids evaluations on how possible changes could influence the consumptions trends. In this perspective, the results of the SFN model and the related simulation analysis may be a useful tool to study and forecast the technology trends in the following years, by analyzing different market scenarios based on well-defined policies. In our paper, the most important result deriving from the simulation exercise is the emphasis on the central role of social forces and demand dynamics in renewable energies adoption: both increasing renewable energy incentives and restricting coal and natural gas use would have accelerated the decarbonization in Germany and the energy transition towards renewable sources. Comparing Figures 6 and 7, Policy 2 appears to be more promising and conducive to an expansion of renewables. The analysis of the SFN model suggests that this occurs because Policy 2 affects connectors representing causal relationships observed in many of the feedback loops compared to Policy 1.

A limitation of the analysis proposed in this paper is the lack of a forecasting exercise on future evolution of this market, with a specific attention to renewables. Indeed, forecasting the diffusion of renewables is critical for planning a suitable energy agenda and setting achievable targets in terms of electricity generation, although the available time series are typically short and pose some difficulty in modeling [50]. A further development of this work should certainly try to employ the UCTT-SFN model for scenario building and forecasting, by also extending its application to a wide number of countries. This could offer a larger perspective on decarbonization and sustainability aims, by accounting for the uncertainties and perturbations to energy systems occurring in the upcoming years.

**Author Contributions:** A.S.: Term, conceptualization, software, investigation, data curation, writing, visualization. L.D.G.: Term, conceptualization, software, validation, investigation, writing, supervision. M.G.: Term, conceptualization, validation, investigation, writing, supervision. All authors have read and agreed to the published version of the manuscript.

**Funding:** This work was partially supported by the Interdepartmental Centre for Energy Economics and Technology "Giorgio Levi Cases", Padua, Italy, under the interdisciplinary project VASE (Valorisation of Agri-food Wastes for Sustainable Energy Production).

**Institutional Review Board Statement:** Not applicable.

**Informed Consent Statement:** Not applicable.

**Data Availability Statement:** The data employed in this paper are available at https://www.bp.com.

**Conflicts of Interest:** The authors declare no conflict of interest.

## Abbreviations

The following abbreviations are used in this manuscript:

| | |
|---|---|
| EEG | Renewable Energy Act |
| EJ | Exajoule |
| IRENA | International Renewable Energy Agency |
| MTI | Massachusetts Institute of Technology |
| NLS | Nonlinear Least Square |
| ODE | Ordinary Differential Equations |
| RETs | Renewable Energy Technologies |
| SFN | Stock-Flow Networks |
| UCTT | Unbalanced Competition among Three Technologies |
| UCRCD | Unbalanced Competition Regime Change Diachronic |

## Appendix A

Appendix A presents a comparison between the model UCTT and the 3PM model proposed by [21]. The 3PM model, which has been the starting point for the UCTT model development, is characterized by the following ODE structure.

$$
\begin{aligned}
z_1'(t) &= m\{[p_{1\alpha} + (q_{1\alpha} + \delta_\alpha)\frac{z_1(t)}{m} + q_{1\alpha}\frac{z_2(t)}{m}](1 - I_{t>c_2}) \\
&+ [p_{1\beta} + (q_{1\beta} + \delta_\beta)\frac{z_1(t)}{m} + q_{1\beta}\frac{z_2(t) + z_3(t)}{m}]I_{t>c_2}\}[1 - \frac{z(t)}{m}]x_1(t) \\
z_2'(t) &= m\{[p_{2\alpha} + (q_{2\alpha} - \delta_\alpha)\frac{z_1(t)}{m} + q_{2\alpha}\frac{z_2(t)}{m}](1 - I_{t>c_2}) + \\
&+ [p_{2\beta} + (q_{2\beta} + \delta_\beta)\frac{z_2(t)}{m} + q_{2\beta}\frac{z_1(t) + z_3(t)}{m}]I_{t>c_2}\}[1 - \frac{z(t)}{m}]x_2(t) \quad \text{(A1)} \\
z_3'(t) &= m\{[p_3 + (q_3 - \delta_\beta)\frac{z_1(t) + z_2(t)}{m} + q_3\frac{z_3(t)}{m}]I_{t>c_2}\}[1 - \frac{z(t)}{m}]x_3(t)
\end{aligned}
$$

$$
\begin{aligned}
m &= m_\alpha(1 - I_{t>c_2}) + m_\beta I_{t>c_2} \\
z(t) &= z_1(t) + z_2(t) + z_3(t)I_{t>c_2}
\end{aligned}
$$

Despite a clear similarity between the 3PM and the UCTT model, there are also some crucial differences. In both phases, the 3PM structure is characterized by a constant dynamic that influences the competitors without distinctions through the parameters $\delta_\alpha$ and $\delta_\beta$. Unlike the 3PM model, the UCTT model is based on the assumption that cross-influence can lead to effects of different magnitudes that are not equally divided between the three products. Specifically, regarding the phase where all the three technologies are present, the two models differ in this:

- 3PM model uses $\delta_\beta$ as a constant hyperparameter to determine the cross-influence in all the equations;
- UCTT model uses $\zeta, \rho$, and $\xi$ to distinguish the effects of the cross-influence effect on each of the three technologies.

In this sense, the UCTT model may be seen as a generalization of the 3PM model. If the restriction $\zeta = \rho = \xi$ applies on the UCCT model, that leads to the 3PM model. It is also noteworthy that in the Appendix of [21], a larger model without any restrictions is presented. The equations defining the second phase of this more complex model are described through multiple parameters, implying a larger generalization of the 3PM and UCTT models.

## Appendix B

Appendix B is devoted to a deeper interpretation of the *internal* and *cross-influence* coefficients of the UCTT model in the Germany energy context. To this end, Figure A1 displays the three relationships that have been modeled through the UCTT: specifically, the first graph represents the trajectory of renewables compared with that of coal and natural gas together; the second shows natural gas versus coal and renewables; the third displays coal versus natural gas and renewables jointly considered.

In the *double-competition phase* of the UCTT, the *cross-influence coefficient* refers to the combined influence of two technologies over the third. In this sense, the UCTT provides a general market analysis, where it is possible to understand how each technology is affected by the combined presence of competitors, while it does not study the reverse, i.e., how a single technology affects a couple of competitors (this is a significant difference compared to the UCRCD model, which allows to estimate and interpret the effect of a technology on the other and viceversa).

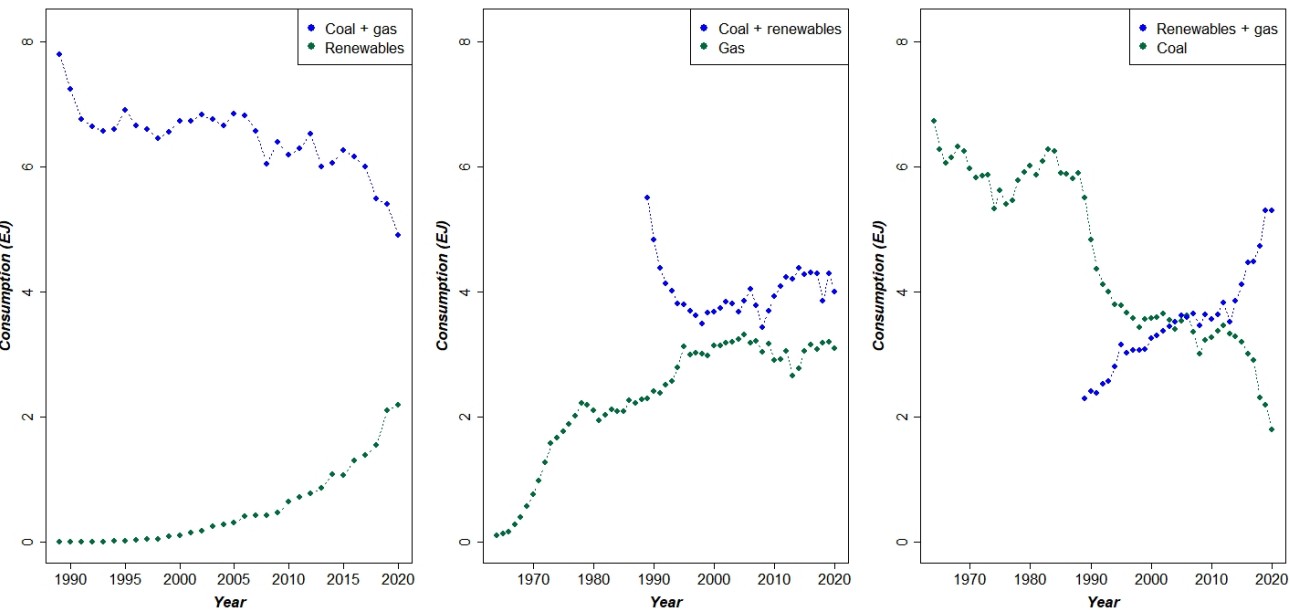

**Figure A1.** Relationships between energy sources as studied in the UCTT model.

By inspecting the first panel in Figure A1, one may observe an almost exponential trend for renewables, consistent with the *internal-influence coefficient* estimated, which is positive and relatively high. On the other hand, the cross-influence of the joint coal and gas series on renewables shows how the "global market" influences their entrance. In this case, the negative but not exceptionally high *cross-influence coefficient* indicates that the market is not yet ready to fully embrace renewables, as the presence of coal and gas limits, albeit in a relatively slight way, their diffusion. The model results suggest that the diffusion of renewables is supported and well-sustained, but it is held back by the *competition* of the other two technologies that still have a dominant presence in the market.

Regarding the relationship between gas and coal-renewables displayed in second panel of Figure A1, the negative *internal-influence coefficient* of the natural gas is not significant, and the parameter $\rho$ is significant at a level of 10%, thus the positive *cross-influence coefficient* does not have a strong significance. This result probably depends on comparing the natural gas stable trend in this phase and a series that, after an initial shock, similarly has a quite stable trajectory.

The third panel in Figure A1 depicts the time series of coal consumption and that formed by natural gas and renewables. The UCTT model outlines a negative *internal-influence coefficient* that entails the declining trend of the series. Instead, the positive

*cross-influence coefficient* emphasizes a support for the individual series by the "global market". The joint trajectory of natural gas and renewables does not have a rapid growth, except for the last observations; it has a considerable slight growth period between 1995 and 2015, in contrast to coal's slightly downward trend. The two evident collapses of coal consumption (the initial and the final one in the *double-competition phase*) may be not directly attributable to the joint influence of competitors, but rather to restrictions enacted by the German government in order to reduce consumption of coal (emphasized through the negative internal influence). This is coherent with the positive effect exerted by natural gas and renewables on coal.

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
