# Peer review of "Modelling Energy Transition in Germany: An Analysis through Ordinary Differential Equations and System Dynamics"

_forecasting, doi:10.3390/forecast4020025_

Round 1

Reviewer 1 Report

The paper proposed by Savio, De Giovanni, and Guidolin evaluates the use of a multivariate diffusion model based on Ordinary Differential Equations to study the dynamic interplay of coal, natural gas, and renewables of energy production in Germany in the period 1964-2020.

Overall, the written work is clear; I had no difficulty in reading and understanding the paper concepts and analysis.

The methods are appropriate to the research aim, studying whether renewables have sufficient competitive strengths towards fossil fuels in Germany. As suggested below, minor issues can be improved in my opinion.

Suggestions for author:

  • The reference to article [1] "The German energy…" can be further developed. I would consider making explicit similarities in methodology and analysis. Furthermore, I would consider comparing the results in the Conclusion, also considering the transition from nuclear energy to renewables.
  • If possible, I would standardize variables' names between the ODE and SDF models (see table 2).
  • In figure 3, I would change the legend's layout, replacing the dots of predicted variables with continuous lines.
  • In table 3, I would add the p-value significance levels, highlighting significant results. 
  • About figure 4, I would add a legend or a description of arrows (also considering yellow, black, purple, orange arrows).
  • In my opinion, Table 5 is not very clear. I would change the layout and add loops' descriptions.
  • In figures 5, 6, and 7, I would add axis names in graphs.
  • In the "Conclusion" section, I would further emphasize the contribution of the SDF model to the overall analysis.

Author Response

We would like to thank Reviewer #1 for her/his valuable comments and suggestions to revise our paper forecasting-1623730. All suggestions have been carefully considered and implemented in the revised version of the paper. On the PDF attached we answer point-by-point to all the questions raised. 

Reviewer 2 Report

This paper presents a System Dynamics model based on the Stock-Flow Network (SFN) describing the energy transition from fossil energy to renewable energy sources in Germany. The model uses ordinary differential equations for diffusion of innovations and describing the interactions between different energy technologies. The model building utilizes the annual data over long periods for Germany. It could be used in testing different long-term policy scenarios of the green energy transition that is now more topical than ever.

Comments:

Contributions: (1) The problem definition and contributions of the paper are scattered in different places in Introduction, They should be collected at the end of Introduction (before the paragraph starting with “The rest of the paper …”. Otherwise, Introduction is a good motivator for the paper.

Results: (2) The methodological part is OK. Of course, we could discuss, if both Figures 2 and 4 are necessary, and if the whole Section 2 could be included in Introduction and Section 3. The main results are given as simulating the alternative scenarios in history. This has some importance in showing that the model works and gives reasonable results. However, it would be more important to tell, how this system can be used in testing some future scenarios; e.g. the existing target to get rid of gas in electricity production. This should be at least commented in Section 6 (see also my comment #3). Another interesting practical point is the decreasing role of nuclear energy. How should it be compensated for?

Conclusions: (3) Conclusion should sum up the results of the paper. Now, it speaks too much about the starting point of the research and omits the actual results. There is one interesting comment. Concerning with the lack of actual forecasting case in this paper, the authors say that the German program was implemented in 2019, and there is only one year’s history available. It is also disturbed by COVID pandemic, and does not make reliable forecasting possible. The question arises how long undisturbed history we need to form scenarios and test them with this tool? I think that the disturbance we now have will last several years, but the decisions (and scenarios) must be done is spite of it. We have to remember that there have been considerable disturbances to our energy economy since 1964 when the time series in this paper start. So, this is a very weak explanation for the lack of the actual case. I would replace this with a short note on the further research topics and discuss the usefulness of this approach in Section 6.

Author Response

We would like to thank Reviewer #2 for her/his valuable comments and suggestions to revise our paper “forecasting-1623730”. All suggestions have been carefully considered and implemented in the revised version of the paper. On the PDF attached we answer point-by-point to all the questions raised. 

Round 2

Reviewer 2 Report

This is the revised version of the paper that I reviewed earlier as an original manuscript. I see that the authors have succeeded in answering the critic I presented in my earlier statement, and I can recommend the publication of this paper. I hope that I will see this system used in studying the actual future scenarios in this very challenging situation we have now.